# Is Metabolic Syndrome Useful for Identifying Youths with Obesity at Risk for NAFLD?

**DOI:** 10.3390/children10020233

**Published:** 2023-01-28

**Authors:** Procolo Di Bonito, Anna Di Sessa, Maria Rosaria Licenziati, Domenico Corica, Malgorzata Wasniewska, Giuseppina Rosaria Umano, Anita Morandi, Claudio Maffeis, Maria Felicia Faienza, Enza Mozzillo, Valeria Calcaterra, Francesca Franco, Giulio Maltoni, Giuliana Valerio

**Affiliations:** 1Department of Internal Medicine, “S. Maria delle Grazie” Hospital, 80078 Pozzuoli, Italy; 2Department of Woman, Child and of General and Specialized Surgery, University of Campania “Luigi Vanvitelli”, 80138 Napoli, Italy; 3Neuro-Endocrine Diseases and Obesity Unit, Department of Neurosciences, Santobono-Pausilipon Children’s Hospital, 80122 Napoli, Italy; 4Department of Human Pathology in Adulthood and Childhood, University of Messina, 98122 Messina, Italy; 5Department of Surgery, Dentistry, Pediatrics and Gynecology, Section of Pediatric Diabetes and Metabolism, University and Azienda Ospedaliera Universitaria Integrata of Verona, 37134 Verona, Italy; 6Department of Precision and Regenerative Medicine and Ionian Area, University of Bari “Aldo Moro”, 70124 Bari, Italy; 7Department of Translational Medical Science, Section of Pediatrics, Regional Center of Pediatric Diabetes, University of Naples “Federico II”, 80131 Napoli, Italy; 8Pediatric Department, Buzzi Children’s Hospital, 20154 Milano, Italy; 9Department of Internal Medicine, University of Pavia, 27100 Pavia, Italy; 10Pediatric Department, Azienda Sanitaria Universitaria Friuli Centrale, Hospital of Udine, 33100 Udine, Italy; 11Pediatric Unit, IRCCS Azienda Ospedaliero-Universitaria di Bologna, 40138 Bologna, Italy; 12Department of Movement Sciences and Wellbeing, University of Napoli “Parthenope”, 80133 Napoli, Italy

**Keywords:** metabolic syndrome, non-alcoholic fatty liver disease, pediatric obesity, insulin-resistance, abdominal obesity

## Abstract

The definition of metabolic syndrome (MetS) in childhood is controversial. Recently, a modified version of the International Diabetes Federation (IDF) definition was proposed using reference data from an international population for high waist circumference (WC) and blood pressure (BP), while the fixed cutoffs for lipids and glucose were not changed. We analyzed MetS prevalence using this modified definition (MetS-IDF_m_) and its association with non-alcoholic fatty liver disease (NAFLD) in 1057 youths (age 6–17 years) with overweight/obesity (OW/OB). A comparison with another modified definition of MetS according to the Adult Treatment Panel III (MetS-ATPIII_m_) was performed. The prevalence of MetS-IDF_m_ was 27.8% and 28.9% by MetS-ATPIII_m_. The Odds (95% Confidence Intervals) of NAFLD was 2.70 (1.30–5.60) (*p* = 0.008) for high WC, 1.68 (1.25–2.26)(*p* = 0.001) for MetS, 1.54 (1.12–2.11)(*p* = 0.007) for low HDL-Cholesterol, 1.49 (1.04–2.13)(*p* = 0.032) for high triglycerides and 1.37 (1.03–1.82)(*p* = 0.033) for high BP. No substantial difference was found in the prevalence of MetS-IDF_m_ and frequency of NAFLD compared to Mets-ATPIII_m_ definition. Our data demonstrate that one third of youths with OW/OB have MetS, whichever was the criterion. Neither definition was superior to some of their components in identifying youths with OW/OB at risk for NAFLD.

## 1. Introduction

The term “metabolic syndrome” (MetS) was proposed in adulthood more than twenty years ago to identify a cluster of several cardiometabolic risk factors (CMRFs) sharing a common pathophysiological “milieu” represented by insulin-resistance (IR). MetS was originally defined by the Adult Treatment Panel III (MetS-ATPIII) as the presence of at least three of the five following factors: abdominal adiposity (expressed as high waist circumference (WC)), high blood pressure (BP), high triglycerides (TG), low high density lipoprotein-cholesterol(HDL-C), and prediabetes/type 2 diabetes [1]. Given the worldwide heterogeneity of abdominal adiposity, in 2006 a statement of the International Diabetes Federation (IDF) recommended the adoption of an alternative definition based on abdominal adiposity (with different cut-offs by ethnicity) plus at least two factors among high BP, high TG, low HDL-C, and hyperglycemia [2].

For several years, the MetS has been considered a useful entity to identify adults at high risk of cardiovascular events or diabetes [3,4]. However, over the time its low clinical utility in adulthood has been demonstrated compared to the single components of MetS [5,6] and subsequently it has been neglected [7].

In children with overweight or obesity (OW/OB), the MetS is still considered a useful tool to recognize individuals with an unhealthy metabolic phenotype characterized by the CMRFs included in the definition of MetS [8,9]. In addition, several studies supported the association of MetS with preclinical atherosclerosis [10], non-alcoholic fatty liver disease (NAFLD) [11], and cardiac impairment [12].

However, in contrast with two main definitions of MetS in adults, at least eleven different definitions have been proposed in children [13] with consequent heterogeneity in the prevalence and variable relationships between MetS and other CMRFs or preclinical signs of organ damage.

This heterogeneity might be ascribed to the adoption of country specific cut-offs in the different pediatric MetS definitions to characterize their single components, especially for WC and BP [13]. Therefore, very recently, Zong et al. in the attempt to unify all the previous definitions, proposed a modified definition [14] based on new pediatric cut-offs of WC and BP. This definition was based on 90th percentile of age- and sex-specific cut-offs of WC obtained in a very large international population [15] plus two criteria among 90th percentile for age- sex- and height of BP obtained in 52,636 non-overweight children [16], and fixed cut-offs for TG, HDL-C, and fasting glucose as proposed by the Expert Panel [17]. However, this modified definition has been not yet validated in youths with OW/OB.

Noteworthy, both MetS and NAFLD have been recognized as growing medical issues worldwide, starting from childhood [18,19,20]. Moreover, robust evidence has strongly linked MetS to NAFLD [18,21,22]; however, whether this modified definition of MetS is effective to identify young people at high risk for NAFLD is equally unexplored.

Therefore, this study aimed to evaluate the prevalence of MetS using the definition based on modified IDF criteria (Mets-IDF_m_) in a large sample of Caucasian youths with OW/OB, compared to a modified definition based on ATPIII criteria (MetS-ATPIII_m_). In addition, it was explored whether this definition is more useful than its single components in identifying individuals with NAFLD.

## 2. Materials and Methods

### 2.1. Subjects

This retrospective study included a sample of 1057 children and adolescents (515 boys and 542 girls) aged 6–17 years consecutively observed in the period June 2016–June 2020 in nine tertiary Italian centers for diagnosis and care of pediatric obesity, as elsewhere described [23].

The records of youths with complete anthropometric, biochemical and abdominal ultrasound data were analyzed. Secondary obesity, previous diagnosis of diabetes, hypertension, liver diseases or any pharmacological treatment were considered as exclusion criteria.

The study was conducted according to the guidelines of the Declaration of Helsinki, and was approved by the Ethics Committee of the AORN Santobono-Pausilipon (Reference number 22877/2020). Informed consent was obtained from the parents or tutors of all participants.

### 2.2. Anthropometric, Biochemical and Ultrasound Examination

All anthropometric data were collected using a standard procedure in each center by an expert examiner. Body Mass Index (BMI) was calculated as weight/height^2^ and subsequently was transformed into standard deviation score (SDS), based upon the Italian BMI percentiles [24]. WC was measured in standing position, midway between the lowest rib and the superior border of the iliac crest using a non-extensible steel tape. Systolic and diastolic BP were measured using standard procedure, as elsewhere described [23]. 

After 12 h of fasting, blood samples were drawn for measurements of glucose, insulin, and lipids. Biochemical data were assessed in the centralized laboratory of each center. Conventional ultrasound evaluation of the liver was performed by one trained radiologist in each center using standard methods [21]. All laboratories belong to the Italian National Health System and are certified according to International Standards ISO9000 (www.iso9000. it accessed on 11 January 2023), undergoing to semi-annual quality controls and inter-lab comparisons, as elsewhere described [21,23].

### 2.3. Definitions

Overweight or obesity were defined by the pediatric Italian curve of BMI [24]. Prepubertal stage was defined by Tanner stage I. 

IR was estimated by the 97th percentile of HOMA-IR distribution by age and gender in normal weight Italian children, as elsewhere described [23].

The modified definition of MetS based on IDF criteria (MetS-IDF_m_) was represented by high WC based on 90th percentile of sex- and age-specific international WC references [15] plus two factors among: (a) high BP (systolic BP ≥ 90th percentile and/or diastolic BP ≥ 90th percentile (sex-, age and height-specific, international BP references)) [16], (b) TG ≥ 100 mg/dL in children aged <10 years or ≥130 mg/dL in children aged ≥10 years, (c) HDL-C < 40 mg/dL or d) impaired fasting glucose (IFG) (fasting glucose ≥ 100 mg/dL) [17].

A modified definition of MetS based on ATPIII criteria (MetS-ATPIII_m_) was used for comparison. MetS-ATPIII_m_ was defined by the presence of at least 3 risk factors among: (a) high WC, (b) high TG and (c) low HDL-C (both defined using the age- and gender-specific cut-offs intercepting at 18 years the ATPIII cut-points) [25] and (d) elevated BP defined as BP ≥ 90th percentile for age, gender and height in children or ≥120/80 in adolescents according to the most recent Clinical Practice Guidelines (CPG) [26]; (e) IFG, as elsewhere described [21].

Assessment of NAFLD was based on the presence of increased echogenicity (brightness) of the liver as compared to the renal cortex. NAFLD was assessed as present or absent, as previously reported [18].

### 2.4. Statistical Analysis

Data were expressed as mean ± SD, or proportions (%). Given the skewed distribution of TG, the statistical analysis of this variable was applied after log-transformation and expressed as median and interquartile range. Chi-square or Fisher’s exact test, as appropriate, were used to compare proportions. Comparisons between groups were performed using the Student’s *t*-test. To analyze the performance of MetS or its factors in the discrimination of the NAFLD, we calculated sensitivity, specificity, positive and negative predictive value by 2 × 2 tables for both definitions. Subsequently, we performed logistic regression analyses using NAFLD as dependent variable and centers, age, prepubertal stage and the definitions of MetS or in alternative of their single components as covariates. A *p* value < 0.05 was considered statistically significant. The statistical analysis was performed using the IBM SPSS Statistics, Version 20.0. Armonk, NY, USA.

## 3. Results

Table 1 reports the main characteristics of youths fulfilling or not MetS criteria using the two definitions. The MetS-IDF_m_ was observed in 294 (27.9%) out of 1057 children and adolescents included in this study. The MetS-ATPIII_m_ was observed in 315 young people (29.8%). There were no differences in the frequency of MetS by sex (MetS-IDFm: boys 28.9%, girls 26.8%, *p* = 0.429; Mets-ATPIIIm: boys 29.1%, girls 30.4%, *p* = 0.640). Patients classified as having MetS, whichever definition was used, were older and more likely to be pubertal, and showed higher values of the remaining variables, except for cholesterol, compared to those without MetS. A significantly higher number of youths with IR was found both in the MetS-IDF_m_ and MetS-ATPIII_m_ groups. 

The frequency of NAFLD was 55.8% in youths with MetS-IDF_m_ and 54.3% in those with MetS-ATPIII_m_ (*p* = 0.336). According to the absence (*n* = 359) or presence (*n* = 698) of insulin-resistance, the proportion of youths with MetS-IDFm, MetS-ATPIIIm and NAFLD was, respectively: 63 (17.5%) vs. 231 (33.1%), 73 (20.3%) vs. 242 (34.7%) and 126 (35.1%) vs. 374 (53.6%) (*p* < 0.0001 for all comparisons).

The percentage of NAFLD among youths with MetS-IDF_m_, MetS-ATPIII_m_, or each altered component of MetS (high WC, high BP, high TG, low HDL-C) defined according the two different criteria, and IFG is represented in Figure 1.

A similar frequency of NAFLD was observed in youths with either MetS-IDF_m_ (55.8%) or MetS-ATPIII_m_ (54.3%) (*p* = 0.336).

Table 2 shows the sensitivity, specificity, positive and negative predictive value of MetS-IDF_m_, and their individual factors (top panel) and MetS-ATPIII_m_ (bottom panel) relative to NAFLD. A higher sensitivity for NAFLD was observed for high WC, high BP using IDFm criteria, while using ATPIII_m_ criteria the best sensitivity was observed for high WC, Low-HDL-C and high BP. Both MetS definitions and their individual components showed weak positive and negative predictive values. 

The Odds of NAFLD for MetS or their single or clustered components is shown in Table 3. According to the MetS-IDF_m_ definition (left panel), high WC had the highest OR, followed by MetS and low HDL-C, independently of centers, age, and prepubertal stage. According to the ATPIII_m_ definition (right panel), high WC had the highest OR, followed by high BP and MetS. At difference with MetS-IDF_m_, neither the association with high TG or low HDL was significant. The OR of NAFLD for MetS was slightly attenuated, but still significant, when IR was added to the model (MetS-IDFm 1.51, 95% CI 1.11–2.04, *p* = 0.008; Mets ATP III 1.36, 95% CI 1.01–1.82, *p* = 0.042). Regarding the cluster of factors included in the MetS-IDF_m_ or MetS-ATPIII_m_ definition, the significant association with NAFLD was observed when at least two factors were combined. Since only one subject showed the cluster of five factors for either classification, it was not possible to calculate the OR for NAFLD for this category.

## 4. Discussion

The present study provides evidence that at least one third of young people with OW/OB have MetS according to the modified definition proposed by Zhong et al. [14]. No substantial difference was found in the comparison with another classification system of MetS, i.e., Mets-ATPIII_m_. In addition, neither definition was superior to its main components, in identifying youths with OW/OB at risk of NAFLD, particularly high WC and low HDL-C for Mets-IDF_m_ and high WC and high BP for MetS-ATPIII_m_. 

To date, the persistence of Mets diagnosed in children with OW/OB is still under debate [27,28]. To complicate matters, the prognostic role of MetS in adults as risk factor for cardiovascular events or diabetes has been recently neglected [6,7]. Consequently, the usefulness of MetS diagnosis in childhood is currently questionable. In fact, since 2009, the American Heart Association suggested to examine tracking and interactions of single CMRF rather than MetS due to its clinical heterogeneity [13]. This proposal has been subsequently reinforced in 2017 by the American Academy of Pediatrics (Committee on nutrition, section on endocrinology on obesity) [29] that emphasized the usefulness to focusing on children with CMRF clustering rather than MetS. 

The main concern about pediatric MetS is represented by its heterogeneous prevalence, as a potential result of the adoption of regional or national WC cut-offs obtained from databases often including children with OW/OB [13,25]. In order to overcome this limitation, Xi et al. proposed WC cutoffs age- and gender-related based on a large international population where young people with OW/OB were excluded [15]. Therefore, the proposal of the definition of MetS based of the 90th percentile of WC derived from that study might to be reasonable. Similarly, cutoffs for BP were taken from seven large nationally representative cross-sectional survey. However, we demonstrated that despite the different cutoffs used to define the five components of the MetS, the frequency of MetS was quite similar in youths with OW/OB as well as the frequency of NAFLD did not differ between MetS-IDF_m_ and MetS-ATPIII_m_^.^

Our findings support the close association among high WC, as surrogate of visceral obesity, NAFLD, and CMRFs as previously reported [12,30,31]. From a pathogenic perspective, an intriguing link of abdominal fat with obesity-related metabolic impairment has been found [32,33]. As a consequence of the shared pathophysiological pathways including IR, hepatic fat has been also associated with CMRFs [22,32]. The tangled pathogenic bases of this association are still not fully explored but a key role for IR in systemic metabolic impairments has been widely supposed [32]. More, fatty liver has been implied in systemic IR development by affecting insulin sensitivity, but contrasting evidence has emerged on the causal link between these two conditions [31]. However, the central regulatory role of IR in several metabolic pathways should be kept in mind [31], as supported by its pathophysiological involvement in the relationship of obesity with NAFLD, prediabetes/diabetes, and cardiovascular outcomes [22,28,32]. In this perspective, our data confirm the higher frequency of youths with IR in the group with MetS, independently of the used definition. 

Worthy of note, the pivotal role of dysmetabolism in NAFLD pathogenesis has been recently underlined [32,33]. In 2020, an international expert panel proposed to rename NAFLD as metabolic (dysfunction)-associated fatty liver disease (MAFLD) in order to provide a better pathophysiological definition of the disease [34]. In fact, several lines of evidence have supported the key role of the tangled dynamics of fatty liver with a wider spectrum of metabolic dysregulation [31,32,34]. Besides the well-known liver-related impairments, MAFLD has been also closely related to different extrahepatic outcomes (e.g., cardiovascular disease, type 2 diabetes, dyslipidemia, and chronic kidney disease) [35,36]. 

Although challenging in children with obesity for its prognostic implications [37], MAFLD definition does not represent a mere semantic change but deeply highlights the “metabolic” origin of fatty liver [32,33]. More specifically, this new nomenclature of fatty liver might be considered as a pathophysiological mirror of the larger underlying dysmetabolism than that included in any MetS definition [32,38]. Indeed, it provided a better depict of the interplay of fatty liver with IR and inflammation [32,34]. Given also the association of visceral adiposity with liver fat and IR [34,38] our findings seemed to further support the clinical usefulness of few metabolic features in identifying children with NAFLD.

Among the examined biochemical components of MetS, low HDL-C appeared to be more significantly associated with NAFLD only using the IDFm criteria. This finding is in line with previous evidence demonstrating a significant diagnostic value for low HDL in predicting steatosis [39,40]. 

Interestingly, the association of high BP with NAFLD should be also considered. The available literature data have shown an intriguing relationship between high BP and liver steatosis [41,42]. As a pathophysiological link, IR has been implicated but its exact role is still to be elucidated [42,43]. 

To sum up, the modified definition of MetS might potentially underestimate the identification of patients at high CMR. Indeed, any proposed MetS definition included IGF, which is the least prevalent component and has a weaker impact on NAFLD [13]. In fact, our data confirmed that IFG showed no association with NAFLD compared to other traits of MetS (e.g., WC, HDL-C, and BP) closely related to high CMR.

More, Di Bonito et al. [12] previously demonstrated in a small sample of children with OW/OB that MetS defined by criteria based on different age and gender-related cut-offs [28] was not superior to WC to identify high ALT levels (as surrogate of NAFLD) or concentric left ventricular hypertrophy. By confirming our previous observation, we expanded these findings using the modified IDF definition. To reinforce the concept of low clinical utility of MetS in identifying young people at risk of NAFLD, it should be underlined that youths with a healthy metabolic profile (i.e., without any MetS component) had an increased risk of NAFLD spanning from normal weight to obesity or morbid obesity [9]. In addition to previous evidence [12], recent data have strengthened the lower accuracy of MetS as marker of fatty liver compared to specific metabolic features in children with OB [21]. In an attempt to explain the low performance of MetS and its components in NAFLD identification, the prominent role of the genetic background should be underpinned [41,42]. Notably, our findings also showed that the association between NAFLD and MetS was dampened when IR was added in the model, suggesting the potential role of other determinants in modulating this relationship. 

To date, the complex and multifactorial landscape of NAFLD pathophysiology has not yet been fully elucidated. A large body of evidence has demonstrated the role of genetics in NAFLD pathophysiology [44,45]. Indeed, different genetic variants (e.g., the I148M allele of the *Patatin-like phospholipase containing domain 3* (*PNPLA3*), the E167K allele of the *Transmembrane 6 superfamily member 2* (*TM6SF2*), the *hydroxysteroid 17-beta dehydrogenase 13* (*HSD17B13*), the *rs1260326* in the glucokinase regulatory protein (*GCKR*), and the *Membrane bound O-acyltransferase domain containing 7-transmembrane channel-like 4* (*MBOAT7-TMC4*) genes have been mainly linked to NAFLD development and progression. More, the intricate interplay of these risk variants with epigenetic, environmental, and nutritional factors has been also implied in the pathophysiology of NAFLD [44].

Our observation is in line with previous studies supporting the concept that the clustering of factors included in the definitions of MetS may not be superior to the single components in relation to cardiac, vascular, or liver damage [12,46,47]. Moreover, the association with NAFLD was more affected by the type of the clustered factors than their number, as previously demonstrated [30,31]. Therefore, the current role of pediatric MetS as marker of high cardiometabolic risk needs to be reconsidered.

Our study presented some limitations that deserve mention. First, the cross-sectional design and the lack of normal-weight individuals should be acknowledged. More, the inclusion of Caucasian youths with OW/OB is not representative of the overall population. Lastly, conventional ultrasound evaluation of the liver was performed by one trained radiologist in each center. Since this method is operator-dependent, we cannot exclude that the frequency of NAFLD might have been underestimated. Indeed, we should acknowledge that the current gold standard for the diagnosis NAFLD is represented by liver biopsy. However, the use of hepatic ultrasound better reflects daily clinical practice, since liver biopsy is not routinely performed due to its invasiveness and ethical issues in childhood [48,49].

The strength includes the large multicenter cohort of deeply phenotyped youths with OB/OW.

In conclusion, our study demonstrated that despite the attempt to have a more uniform definition based on international standards, the modified MetS definition was not better either than another widely used classification or single components, namely high WC and low HDL-C, with respect to NAFLD. Therefore, our data do not support the superiority of the international criteria of MetS to suspect the NAFLD in youths with OW/OB.

## Figures and Tables

**Figure 1 children-10-00233-f001:**
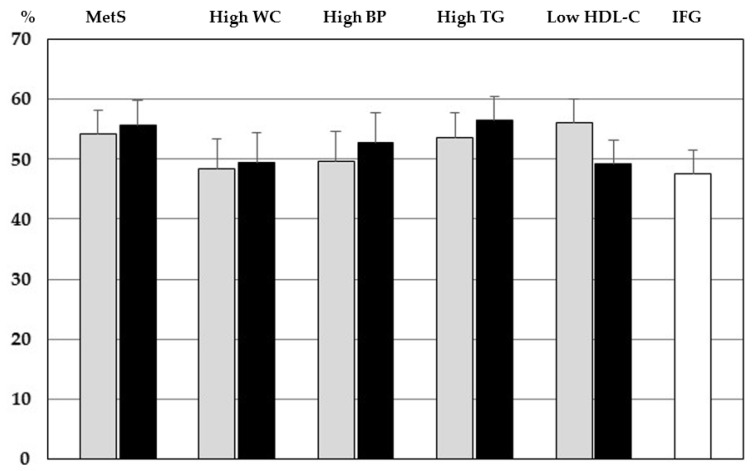
Proportion of youths with NAFLD by MetS-IDF_m_ (grey bars) and MetS-ATPIII_m_ (black bars) or single components. Abbreviations: BP: blood pressure, HDL-C: high density lipoprotein-cholesterol, IFG: impaired fasting glucose, MetS: metabolic syndrome, TG: fasting triglycerides, WC: waist circumference.

**Table 1 children-10-00233-t001:** Characteristics of youths classified by Mets-IDF_m_ or MetS-ATPIII_m_ criteria.

	Mets-IDF_m_ Absent	Mets-IDF_m_ Present	*p* Value	MetS-ATPIII_m_ Absent	MetS-ATPIII_m_ Present	*p* Value
n = 1057, n (%)	763 (72.2)	294 (27.8)		742 (70.2)	315 (29.8)	
Age (years)	11.2 ± 2.5	11.9 ± 2.4	<0.0001	11.2 ± 2.4	12.0 ± 2.5	<0.0001
Prepubertal, *n* (%)	111 (14.5)	28 (9.5)	0.030	109 (14.5)	30 (9.4)	0.023
Female sex, *n* (%)	397 (52.0)	145 (49.3)	0.429	377 (50.8)	165 (52.4)	0.640
BMI, kg/m^2^	30.0 ± 5.1	33.1 ± 6.3	<0.0001	30.0 ± 5.2	33.0 ± 5.8	<0.0001
BMI-SDS	2.2 ± 0.5	2.6 ± 0.6	<0.0001	2.2 ± 0.6	2.6 ± 0.6	<0.0001
WC, cm	92.6 ± 13.2	98.7 ± 13.6	<0.0001	92.0 ± 12.9	99.8 ± 13.6	<0.0001
G_0_, mg/dL	85.1 ± 8.9	91.8 ± 11.2	<0.0001	85.3 ± 9.1	91.0 ± 11.1	<0.0001
I_0_ (µU/mL)	16.3 (11.5–23.8)	21.1 (14.5–31.6)	<0.0001	16.1 (11.5–23.9)	20.6 (13.9–29.5)	<0.0001
HOMA-IR	3.4 (2.3–4.9)	4.8 (3.3–7.1)	<0.0001	3.4 (2.3–4.9)	4.5 (3.2–6.8)	<0.0001
TC, mg/dL	153.3 ± 28.9	154.2 ± 30.9	0.647	153.5 ± 28.5	153.0 ± 29.1	0.706
HDL-C, mg/dL	49.5 ± 9.5	40.8 ± 8.3	<0.0001	49.7 ± 10.2	41.0 ± 6.1	<0.0001
TG, mg/dL	73.0 (59.0–94.0)	105.0 (74.3–130.0)	<0.0001	75.0 (59.0–96.0)	94.0 (70.0–125.0)	<0.0001
SBP, mmHg	109.0 ± 13.8	117.9 ± 10.6	<0.0001	108.1 ± 12.3	119.6 ± 13.0	<0.0001
DBP, mmHg	65.5 ± 9.4	70.5 ± 9.4	<0.0001	65.0 ± 8.8	71.5 ± 10.1	<0.0001
IR, *n* (%)	467 (61.2)	231 (78.6)	<0.001	456 (61.5)	242 (76.8)	<0.0001
NAFLD, *n* (%)	334 (43.8)	164 (55.8)	<0.0001	329 (44.3)	171 (54.3)	0.003

Data are expressed as mean ± SD, *n* (%), median (IQ range). Abbreviations: BMI: body mass index, DBP: diastolic blood pressure, HDL-C: high density lipoprotein-cholesterol, HOMA-IR: homeostasis model assessment, G_0_: fasting glucose, I_0_: fasting insulin, IR: insulin-resistance; MetS-ATPIII m: metabolic syndrome- Adult Treatment Panel III modified, MetS-IDFm: metabolic syndrome-International Diabetes Federation modified, NAFLD: non-alcoholic fatty liver disease, TC: total cholesterol, TG: fasting triglycerides, SBP: systolic blood pressure, WC: waist circumference.

**Table 2 children-10-00233-t002:** Sensitivity, specificity, positive and negative predictive value (95%Cl) of MetS, and their single factors with respect to NAFLD by MetS-IDF_m_ or MetS-ATPIII_m_.

MetS-IDF_m_	Sensitivity	Specificity	PPV	NPV
MetS	0.33 (0.30–0.36)	0.77 (0.74–0.79	0.56 (0.53–0.59)	0.56 (0.53–0.59)
High WC	0.98 (0.97–0.99)	0.07 (0.05–0.08)	0.48 (0.45–0.51)	0.77 (0.74–0.80)
High BP	0.70 (0.67–0.73)	0.36 (0.33–0.39)	0.49 (0.46–0.53)	0.58 (0.55–0.61)
Low HDL-C	0.26 (0.23–0.29)	0.82 (0.80–0.84)	0.56 (0.53–0.59)	0.55 (0.52–0.58)
High TG	0.18 (0.15–0.20)	0.87 (0.84–0.89)	0.54 (0.51–0.57)	0.54 (0.51–0.57)
IFG	0.09 (0.08–0.11)	0.91 (0.89–0.92)	0.48 (0.44–0.51)	0.53 (0.50–0.56)
**MetS-ATPIII_m_**	**Sensitivity**	**Specificity**	**PPV**	**NPV**
MetS	0.34 (0.31–0.37)	0.74 (0.71–0.77)	0.54 (0.51–0.57)	0.56 (0.53–0.59)
High WC	0.90 (0.88–0.92)	0.17 (0.15–0.20)	0.49 (0.46–0.53)	0.66 (0.63–0.68)
High BP	0.42 (0.39–0.45)	0.66 (0.63–0.69)	0.53 (0.50–0.56)	0.56 (0.53–0.59)
Low HDL-C	0.62 (0.59–0.65)	0.43 (0.40–0.46)	0.49 (0.46–0.52)	0.56 (0.52–0.59)
High TG	0.10 (0.09–0.12)	0.93 (0.91–0.94)	0.57 (0.54–0.60)	0.54 (0.51–0.57)
IFG	0.09 (0.08–0.11)	0.91 (0.89–0.92)	0.48 (0.44–0.51)	0.53 (0.50–0.56)

Abbreviations: BP: blood pressure, HDL-C: high density lipoprotein-cholesterol, IFG: impaired fasting glucose, MetS: metabolic syndrome, MetS-ATPIII m: metabolic syndrome-Adult Treatment Panel III modified, MetS-IDFm: metabolic syndrome-International Diabetes Federation modified, NPV: negative predictive value, PPV: positive predictive value, TG: triglycerides, WC: waist circumference.

**Table 3 children-10-00233-t003:** Odds ratio (95%Cl) adjusted for centers, age and prepubertal stage of NAFLD according to the two definitions MetS-IDF_m_ or MetS-ATPIII_m_ and their single or clustered components.

	MetS-IDF_m_		MetS-ATPIII_m_	
	NAFLD	*p* Value	NAFLD	*p* Value
MetS	1.68 (1.25–2.26)	0.001	1.47 (1.10–1.97)	0.009
High WC	2.70 (1.30–5.60)	0.008	2.03 (1.35–3.05)	0.001
High BP	1.37 (1.03–1.82)	0.033	1.53 (1.14–2.05)	0.004
Low HDL-C	1.54 (1.12–2.11)	0.007	1.15 (0.88–1.49)	0.315
High TG	1.49 (1.04–2.13)	0.032	1.55 (0.98–2.46)	0.063
IFG	1.04 (0.65–1.65)	0.878	1.04 (0.65–1.65)	0.878
One factor	2.14 (0.75–6.08)	0.154	1.35 (0.70–2.59)	0.372
Two factors	1.72 (1.24–2.39)	0.001	1.75 (1.31–2.34)	<0.0001
Three factors	1.67 (1.24–2.25)	0.001	1.46 (1.10–1.95)	0.010
Four factors	2.05 (1.05–4.02)	0.036	1.93 (1.04–3.58)	0.036

Abbreviations: BP: blood pressure; HDL-C: high density lipoprotein-cholesterol; IFG: impaired fasting glucose; MetS: metabolic syndrome; MetS-ATPIII m: metabolic syndrome- Adult Treatment Panel III modified, MetS-IDFm: metabolic syndrome-International Diabetes Federation modified, TG: triglycerides; WC: waist circumference.

## Data Availability

The data presented in this study are available on request from the corresponding author. The data are not publicly available due to privacy concerns.

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
