# Peer review of "Is Metabolic Syndrome Useful for Identifying Youths with Obesity at Risk for NAFLD?"

_children, 2023, doi:10.3390/children10020233_

Round 1

Reviewer 1 Report

I read with interest a paper entitled “Is metabolic syndrome useful for identifying youths with obesity at risk for NAFLD?”. 

Although the topic of the paper is interesting and well written, several issues should be addressed.

First, IDF definition was provided in 2006, thus it is not a new definition. Second, association of NAFLD with cardiometabolic risk factors including metabolic syndrome is well known. Third, this study is limited to youths with overweight and obesity. Fourth, NAFLD was assessed with ultrasonography, not biopsy. Thus, I am not convinced that this study added new knowledge.

To be more clinically significant, I recommend following revisions.

Please compare with other definitions of metabolic syndrome such as NCEP ATPIII.

Insulin resistance plays key role in relationship between NAFLD and metabolic syndrome. Thus, please provide fasting insulin, HOMA-IR, and insulin resistance in the results, and show the association between insulin resistance, metabolic syndrome, and NAFLD.

Please compare the participants after subdividing into overweight and obesity.

Please compare the participants after subdividing into male and female.

Please show OR (95% CI) of number of metabolic syndrome component for NAFLD.

Author Response

First, IDF definition was provided in 2006, thus it is not a new definition. Second, association of NAFLD with cardiometabolic risk factors including metabolic syndrome is well known. Third, this study is limited to youths with overweight and obesity. Fourth, NAFLD was assessed with ultrasonography, not biopsy. Thus, I am not convinced that this study added new knowledge.

 To be more clinically significant, I recommend following revisions.

We thank the reviewer for his/her useful comments. We agree that the IDF definition is not new, if we look at the 5 components. The novelty relies upon the reference standard used for WC and BP. Therefore we amended the word “new” throughout the manuscript.

Please compare with other definitions of metabolic syndrome such as NCEP ATPIII.

We added a comparison with NCEP ATPIII definition, consequently the Results and Discussion sections have been implemented with these new results.

Insulin resistance plays key role in relationship between NAFLD and metabolic syndrome. Thus, please provide fasting insulin, HOMA-IR, and insulin resistance in the results, and show the association between insulin resistance, metabolic syndrome, and NAFLD.

Table 1 shows fasting insulin and HOMA-IR across the different groups.

Please compare the participants after subdividing into overweight and obesity.

It was not possible to do this sub-analysis since the % of OW was very small: n =114 (10.8 %)

Please compare the participants after subdividing into male and female.

We provided data about sex differences in the prevalence of MetS, according the two definitions. We added the following sentence in the Results section:  There were no differences in the frequency of MetS by sex (MetS-IDFm: boys 28.9%, girls 26.8%, p =0.429; Mets-ATPIIIm: boys 29.1%, girls 30.4%, p =0.640).

We feel that after the addition of new results comparing two definition systems, further sub analyses by sex might weight the Results section.

Please show OR (95% CI) of number of metabolic syndrome component for NAFLD.

We added this analysis in table 3.

Reviewer 2 Report

The article entitled: "Is metabolic syndrome useful for identifying youths with obesity at risk for NAFLD?" is very interesting.

However there are some aspects to be clarified. I would like to ask the authors to provide more information regarding the performance of ultrasound used for the diagnosis of NAFLD. This is an operator-dependent method, so the authors should provide more information.

Were the ultrasounds always performed by the same operator?

These aspects need to be better addressed by the authors in the study methods.

I suggest discussing these aspects also within the limits of the study.

Author Response

However there are some aspects to be clarified. I would like to ask the authors to provide more information regarding the performance of ultrasound used for the diagnosis of NAFLD. This is an operator-dependent method, so the authors should provide more information.

Were the ultrasounds always performed by the same operator?

These aspects need to be better addressed by the authors in the study methods.

 In the Methods it was specified that “Conventional ultrasound evaluation of the liver was performed by one trained radiologist in each center using standard methods) [18].

We agree with the reviewer that liver ultrasonography is operator-dependent. For this reason, logistic analyses were controlled also by centre.  

I suggest discussing these aspects also within the limits of the study.

We added this sentence in the Limitation section: Lastly, conventional ultrasound evaluation of the liver was performed by one trained radiologist in each center. Since this method is operator-dependent we cannot exclude that the frequency of NAFLD might have been underestimated.

Reviewer 3 Report

I´m very satysfied with article. 

Author Response

we thank the reviewer for his/her positive feedback.

Reviewer 4 Report

Procolo Di Bonito et al demonstrated the prevalence of NAFLD in overweight/obese children/adolescents with or without metabolic syndrome according to the updated criteria and discussed the predictive effects of these new criteria compared to its main components using a retrospective dataset. The study is well-designed and the presentation of results is sound. And they discussed concept is important for clinical understanding in the field and has the potential to guide clinical strategies. I would suggest publication.

To my curiosity, since the authors discussed the “new” diagnostic strategy of MetS in children/adolescents, how are the predictive effects of the “old” MetS criteria compared to its main components? This could be an interesting topic in the discussion section.

Author Response

Procolo Di Bonito et al demonstrated the prevalence of NAFLD in overweight/obese children/adolescents with or without metabolic syndrome according to the updated criteria and discussed the predictive effects of these new criteria compared to its main components using a retrospective dataset. The study is well-designed and the presentation of results is sound. And they discussed concept is important for clinical understanding in the field and has the potential to guide clinical strategies. I would suggest publication.

We thank the reviewer for his/her positive feedback.

To my curiosity, since the authors discussed the “new” diagnostic strategy of MetS in children/adolescents, how are the predictive effects of the “old” MetS criteria compared to its main components? This could be an interesting topic in the discussion section.

As suggested by reviewer 1, we compared the modified new definition (Mets-IDFm) to another modified  definition system which has been widely used in the pediatric literature (MetS-ATPIIIm). Consequently the Results and Discussion sections have been implemented.

Round 2

Reviewer 1 Report

I am pleased with the progress of the manuscript.

I suggest some additional comment for this study. 

1.I think providing references regarding increase in prevalence of  metabolic syndrome and NAFLD among children in introduction can emphasize importance of this study.

2.Insulin resistance plays a key role in relationship between metabolic syndrome and NAFLD. Please provide proportion of metabolic syndrome and NAFLD according to presence of insulin resistance. 

3.Information of liver biopsy was not provided in this study. Please provide this limitation in the discussion section. 

Author Response

We thank again the reviewer and amended the text, as suggested.

1.I think providing references regarding increase in prevalence of  metabolic syndrome and NAFLD among children in introduction can emphasize importance of this study.

We added the following sentence: “Noteworthy, both MetS and NAFLD have been recognized as worlwide growing medical issues, starting from childhood [18-20].” Please, note that references 18-20 are new .

2.Insulin resistance plays a key role in relationship between metabolic syndrome and NAFLD. Please provide proportion of metabolic syndrome and NAFLD according to presence of insulin resistance. 

We made the calculations and the following sentence was added.

According to the absence (n=359) or presence (n= 698) of insulin-resistance, the pro-portion of youths with MetS-IDFm, MetS-ATPIIIm and NAFLD was respectively: 63 (17.5%) vs 231 (33.1%), 73 (20.3%) vs 242 (34.7%) and 126 (35.1%) vs 374 (53.6%) (p < 0.0001 for all comparisons).

3.Information of liver biopsy was not provided in this study. Please provide this limitation in the discussion section. 

In the limitations section we added the following sentence: “Indeed, we should acknowledge that the current gold standard for the diagnosis NAFLD is represented by liver biopsy. However, the use of hepatic ultrasound better reflects daily clinical practice, since liver biopsy is not routinely performed due to its invasiveness and ethical issues in childhood [49, 50].” Please, note that references 49 and 50 are new .

Reviewer 2 Report

I am satisfied with the article

Author Response

Thank you.